# RNA Profile of Cell Bodies and Exosomes Released by Tumorigenic and Non-Tumorigenic Thyroid Cells

**DOI:** 10.3390/ijms25031407

**Published:** 2024-01-24

**Authors:** Valentina Maggisano, Francesca Capriglione, Catia Mio, Stefania Bulotta, Giuseppe Damante, Diego Russo, Marilena Celano

**Affiliations:** 1Department of Health Sciences, University “Magna Graecia” of Catanzaro, 88100 Catanzaro, Italy; vmaggisano@unicz.it (V.M.); francesca.capriglione@studenti.unicz.it (F.C.); bulotta@unicz.it (S.B.); d.russo@unicz.it (D.R.); 2Department of Medical and Biological Sciences, University of Udine, 33100 Udine, Italy; catia.mio@uniud.it (C.M.); giuseppe.damante@uniud.it (G.D.)

**Keywords:** exosomes, thyroid cancer cells, RNAs, therapeutic strategies, transcriptome

## Abstract

Tumor cells release exosomes, extracellular vesicle containing various bioactive molecules such as protein, DNA and RNA. The analysis of RNA molecules packaged in exosomes may provide new potential diagnostic or prognostic tumor biomarkers. The treatment of radioiodine-refractory aggressive thyroid cancer is still an unresolved clinical challenge, and the search for biomarkers that are detectable in early phase of the disease has become a fundamental goal for thyroid cancer research. By using transcriptome analysis, this study aimed to analyze the gene expression profiles of exosomes secreted by a non-tumorigenic thyroid cell line (Nthy-ori 3.1-exo) and a papillary thyroid cancer (TPC-1-exo) cell line, comparing them with those of cell bodies (Nthy-ori 3.1-cells and TPC-1-cells). A total of 9107 transcripts were identified as differentially expressed when comparing TPC-1-exo with TPC-1-cells and 5861 when comparing Nthy-ori 3.1-exo with Nthy-ori 3.1-cells. Among them, *Sialic acid-binding immunoglobulin-like lectins 10 and 11 (SIGLEC10, SIGLEC11)* and *Keratin-associated protein 5 (KRTAP5-3)* transcripts, genes known to be involved in cancer progression, turned out to be up-regulated only in TPC-1-exo. Gene ontology analysis revealed significantly enriched pathways, and only in TPC-1-exo were the differential expressed genes associated with an up-regulation in epigenetic processes. These findings provide a proof of concept that some mRNA species are specifically packaged in tumor-cell-derived exosomes and may constitute a starting point for the identification of new biomarkers for thyroid tumors.

## 1. Introduction

Cell-to-cell communication has been recognized to play a crucial role in carcinogenesis and, in this context, exosomes have been proposed as efficient messengers by carrying genetic and proteomic information among neighboring cells or distant organs [1,2]. Exosomes are small lipid bilayer membrane vesicles of endocytic origin with a size ranging from 30 to 150 nm. They are naturally secreted during both physiological and pathological conditions and their concentrations rise in cancer, potentially to serve functions of growing cancer cells [3,4]. Thanks to their high stability in the bloodstream and intrinsic capacity to deliver various bioactive molecules into recipient cells, exosomes have evoked an increased interest in cancer research as biomarkers, biological targets or drug delivery vehicles. Exosomes derived from tumor cells contains DNA, mRNAs and miRNAs, which affects innate immune responses in tumor microenvironments. Exosome-related gene expression status has been widely associated with tumor progression [5,6]. Therefore, the detection of specific mRNAs in exosomes may be used to detect cancer cells and to evaluate cancer progression [7,8].

Recently, several studies have been focused on the role of exosomes in thyroid cancer (TC), the ninth most common tumor worldwide [9]. This tumor comprises different morphological phenotypes, characterized by an uneven degree of proliferation, differentiation and biological aggressiveness. Different genetic alterations and various signaling pathways have been associated with distinct types of thyroid neoplasms [10]. Papillary thyroid cancer (PTC) represents approximately 80% of all thyroid malignancies and is generally associated with a good prognosis thanks to the available therapeutic tools (surgery followed by radioiodine therapy) [11]. Unfortunately, this approach is not effective for patients with the more aggressive subtypes (10–15%). 

Recently an update on the characterization of the morphological, mutational, transcriptional and genomic profiling of the different types of TC has claimed the relevance of new molecular approaches for a better and useful characterization of these tumors [10]. In this context, transcriptome analysis may have an essential role in deciphering molecular derangements present in cancer, establishing new molecular biomarkers for diseases [12]. In fact, while genome analysis describes a relatively fixed or very low-rate changing situation, transcriptomes are context-dependent and amenable to very fast modifications and therefore more reliable as molecular biomarker.

This study was aimed to investigate the molecular features of PTC cells by analyzing the gene expression profile of exosomes secreted by two thyroid cell lines (non-tumorigenic and tumorigenic). A comparison with the transcriptome derived from cell bodies was performed in order to provide a proof of concept that the analysis of exosomal transcriptomes may identify useful molecular features to characterize thyroid cancer and to ideate new therapeutic approaches. 

## 2. Results

### 2.1. Gene Expression Analysis of Papillary Thyroid Cancer and Non-Tumorigenic Cells

In a previous report, we published results related to the characterization of Nthy-ori 3.1 and TPC-1 cell-derived exosomes by Dynamic Light Scattering (DLS) analysis and Western blotting analysis [13]. From the DLS analysis, we found that exosomes had a size distribution in the expected range of 40–160 nm.

From the Western blotting assay, widely used for the analysis of exosomes, we evaluated the expression of the tetraspanin protein CD63, a classical hallmark of exosomes, published in Maggisano et al., 2022 [13], and as reported in several studies [14,15]. To exclude cell contamination in the exosome lysates, we also evaluated the expression of the protein calregulin. CD63 and calregulin were detected only in the extracted exosomes and in cells, respectively.

A high-throughput RNA sequencing analysis was performed to assess the variation of the gene expression profile between cell bodies and exosomes of papillary (TPC-1-cells and TPC-1-exo, respectively) and non-tumorigenic (Nthy-ori 3.1-cells and Nthy-ori 3.1-exo, respectively) thyroid cells. A mean of 6.8 million reads/sample were generated, and the expression of 20,813 genes was detected. Principal component analysis (PCA) used to analyze the distribution and uniformity of the independent samples, which revealed a clear separation in the overall expression distribution between cells and exosomes-derived mRNAs (Figure 1a). Interestingly, distances between cells and exosomes are larger than the distance between the two cell types, TPC-1 and Nthy-ori 3.1. 

The expression of 15,844 and 17,623 genes was detected in TPC-1-cells and TPC-1-exo RNAs, respectively, while 16,233 and 18,931 genes were identified in Nthy-ori 3.1-cells and Nthy-ori 3.1-exo RNAs, respectively. Following our analysis, we observed that 13,659 genes were expressed in both TPC-1-cells and TPC-1-exo, and that 15,580 genes were commonly expressed in Nthy-ori-3.1-cells and Nthy-ori 3.1-exo. In addition, 2185 RNAs were detected only in TPC-1-cells, and 653 RNAs were detected in Nthy-ori 3.1-cells, while 3964 RNAs and 3351 RNAs were detected only in exosomes secreted by TPC-1 and Nthy-ori 3.1, respectively (Figure 1b). These results suggest the existence of selective mechanisms that drive RNA packing in exosomes.

### 2.2. Transcriptomic Differences between Papillary Thyroid Cancer and Non-Tumorigenic Cells

Applying the criteria of fold change ≥4 or ≤4 and adjusted *p*-value < 0.05 (FDR *p*-value), differential expressed genes (DEGs) between cells and exosomes were identified. A total of 9107 DEGs were assessed when comparing TPC-1-exo with TPC-1-cells, while 5861 DEGs were evaluated when comparing Nthy-ori 3.1-exo with Nthy-ori 3.1 cells. 

In Figure 2 and Figure 3, volcano plots and hierarchical clustering show the variation of mRNA expression between cells and exosomes. Specifically, 2574 RNAs were up-regulated in TPC-1-exo compared to cell-derived RNAs, while 6533 transcripts turned to be down-regulated (Figure 2a). The heatmap of this experiment shows two neatly distinct transcripts clusters, indeed, in exosomes, and cell bodies’ low expression is depicted in blue, while their high expression is depicted in red (Figure 2b). 

For the other cell line, 2885 RNAs were up-regulated in Nthy-ori 3.1 exosomes compared to cell-derived RNAs, while 2976 transcripts turned out to be down-regulated (Figure 3a). The heatmap of the Nthy-ori 3.1 cell line clearly indicates that most transcripts were up-regulated in exosomes compared to cell bodies (Figure 3b).

In order to outline the molecular pathways mostly represented in TPC-1-exo and Nthy-ori 3.1-exo versus the corresponding cells bodies, differentially expressed transcripts were subjected to gene ontology analysis. The results are shown in Appendix A. For each cell line, differences are observed between exosomes and cell bodies, especially for cancer-related pathways. However, only in TPC-1-exo were DEGs associated with the up-regulation of three biological events involved in epigenetics processes: the negative regulation of gene expression, the repression of transcription by altering the structure of chromatin, and nucleosome assembly. 

Then, we focused on single transcripts, and in Table 1 and Table 2, the top 20 DEG genes in TPC-1 and Nthy-ori 3.1 exosomes compared to cells are summarized, respectively. In TPC-1-exo and Nthy-ori 3.1-exo, eight out of twenty mRNAs were common, suggesting some similarity in sorting mRNAs in vesicles. A long intergenic noncoding RNA, LOC440173, known to be involved in cell growth and the cell survival of kidney cancer [16] but not well characterized in thyroid cancer, resulting in greater up-regulation in TPC-1-exo than in Nthy-ori 3.1-exo. 

Regarding different up-regulated mRNAs, in TPC-1-exo, we observed a high expression of Sialic acid-binding immunoglobulin-like lectins (*SIGLEC10* and *SIGLEC11*) and Keratin-associated protein 5 (*KRTAP5-3*). Some studies analyzed the role of these genes in cancer. In patients with hepatocellular carcinoma, the up-regulation of *SIGLEC10* has been associated with poor survival [17]. In breast cancer cells, *KRTAP5-5* has been identified as a driver for vascular invasion, an important step in the malignant progression toward metastasis [18]. The specific biological roles of these genes in the different subtypes of thyroid cancer are still unclear.

In order to obtain a large amount of preliminary data that can help to better specify the transcript pattern present in exosomes, we compared TPC-1-exo with NThy-ori 3.1-exo. Applying the criteria used for the previous analysis, 4374 DEGs were assessed and, when comparing exo-derived TPC1 and exo-derived NThy-ori 3.1, were found in 283 up- and 4091 down-regulated RNAs (Figure 4). In particular, 51 and 3585 transcripts were expressed only in TPC1-1- and NThy-ori 3.1-derived ones, respectively. 

## 3. Discussion

The incidence of thyroid cancer, the most common endocrine malignancy, is rapidly increased in recent years [19]. On the basis of differentiation stage, histology, and causal mutation, it is subdivided into papillary (PTC), follicular (FTC), poorly differentiated (PDTC), anaplastic (ATC) and medullary (MTC) thyroid cancer [20]. Patients with thyroid cancer are mainly diagnosed with PTC and usually respond to conventional therapies. Among PTC, however, a subgroup of patients develops an advanced disease and resistance to standard treatment: tumor cells acquiring invasive capabilities leading to metastasis formation [21]. Recently, novel therapeutic approaches have been proposed [22], but their success is limited since the mechanisms underlying the development of this aggressive subtype of PTC are not yet well understood.

Exosomes consists of a subpopulation of extracellular vesicles enclosed by a lipid bilayer, which are cell secreted after the fusion of intracellular vesicles with the plasma membrane. Exosomes contain several components to be delivered to target cells (either surrounding or distant). A large body of evidence indicates that exosomes play an important role as carriers for cell-to-cell communication and exchange. Exosomes are able to transfer functionally active biological molecules that can have various effects, from protective to pathologic [15]. Therefore, the evaluation of exosomes released by cells into bodily fluids can provide biological indicators for different diseases of distinct organs including the central nervous system, liver, kidney, lung and arteries [23].

Exosome plays a role in cancer; in particular, it has been reported that tumor cells release exosomes and participate in the development and maintenance of a tumor-promoting microenvironment [24]. Furthermore, these vesicles play distinct functions in the control of tumor growth and progression, tumor invasion, neovascularization, immune escape and metastasis [25]. 

Exosomes can alter the biological phenotypes of neighboring cancer cells by activating the receptors or change RNA or miRNA expression in the host cancer cells. In relation to this class of molecules, several data indicate that tumor-derived exosomes are rich in specific RNA and miRNAs; thus, they could be used as tumor markers. Indeed, a difference of the RNA content has been shown between patients with glioblastoma multiforme and healthy subjects. In particular, exosomes from subjects suffering of glioblastoma have increased levels of EGFRvIII mRNA, which thus can be evaluated for a glioblastoma diagnosis. Moreover, EGFR present in exosome membranes could also represent a possible marker for lung cancer diagnosis [23]. Therefore, exosome evaluation for cancer diagnosis can be included in the growing field of liquid biopsy, avoiding the need of surgical procedures to remove tissue, therefore performing a “solid” biopsy.

Recently, in the thyroid cancer field, extracellular vesicle function has been investigated by treating cells with extracellular vesicles isolated from the plasma or serum of patients and from conditioned culture media [21]. Tumorigenic progression was only correlated with the presence of specific extracellular vesicle components (proteins, miRNAs, lncRNAs or circRNAs); however, no specific evidence indicates that progression was really due to the identified molecular cargo. 

Several studies have analyzed the role of miRNAs circulating in extracellular vesicles as biomarkers in thyroid cancer [26,27,28,29]. Recently, a high expression of miR-31-5p, miR-222-3p and let-7i-3p was found in exosomes isolated from TPC-1 and K1, two thyroid cancer cell lines [13] and extracellular vesicles isolated from thyroid tissue after BRAFV600E activation [30]. Exosomes naturally released from distinct cell types can also have beneficial effects in several human diseases. In fact, it has been shown that exosomes derived from bone marrow mesenchymal stem cells (MSC) play a protective role in experimental models of myocardial ischemia/reperfusion as well as brain damage and hypoxia-induced pulmonary hypertension. By transporting miR-133, MSC-derived exosomes are able to stimulate neurite outgrowth in cultured cells from the nervous system. MSC-derived exosome may also play a protective role in models of liver fibrosis or renal damage [23].

In the present study, differences between mRNA species present in exosomes or cell bodies of two continuous thyroid-derived cell lines were delineated. We have used two cell lines that are completely different in terms of tumorigenicity, a tumorigenic cell line (TPC-1) and a non-tumorigenic one (Nthy-ori 3.1). A schematic protocol of the sample processing is depicted in Appendix A. 

PCA analysis revealed that the difference between exosomal RNAs and cell bodies RNAs is larger compared to the differences between RNAs present in the cell bodies of the two lines. This feature suggests the existence of a significant “selection” for RNA species that are delivered through exosomes. 

Our study analyzed 20,861 genes and showed a large difference in the overall expression distribution between cells and exosomes. Data obtained from the gene expression profiling-based analysis indicate that 9107 genes were differently expressed in TPC-1-exo when comparing TPC-1-cells and 5861 genes in Nthy-ori 3.1-exo. More in detail, 2574 genes were up-regulated in TPC-1-exo, while 6533 were down-regulated. Moreover, by comparing TPC-1-exo with Nthy-ori 3.1-exo, 51 genes were expressed only in TPC-1 and 3585 in NThy-ori 3.1, providing a potential role of exosomal mRNAs in thyroid tumorigenesis.

Our study was focused on delineating, for the first time, the different profile of mRNAs among cell bodies and exosomes released by a tumorigenic and a non-tumorigenic thyroid cell line. Analyzing the pathways in which commonly up- or down-regulated genes are involved, we found, only in TPC-1-exo, the high expression of three biological processes: the negative regulation of gene expression, the repression of transcription by alteration of chromatin structure, and nucleosome assembly. All of these processes are involved in epigenetic regulation.

Our findings clearly indicate that transcript patterns present in exosomes are different to those present in cell bodies, suggesting a specific control of RNA inclusion in exosomes or a RNA-based selection of released microvesicles. Interestingly, a significant fraction of the top 20 up-regulated RNA in exosomes compared to RNA present in cell bodies is common in the two investigated cell lines, suggesting that RNA inclusion in exosomes is at least partially conserved among cell types with different tumorigenic potential. In terms of biopathological meaning, among the RNAs whose amount is different between exosomal transcript of the two cell lines of interest are *SIGLEC10*, *SIGLEC11* and *KRTAP5-3*. *SIGLEC* genes encodes for sialic-acid-binding immunoglobulin-like lectins proteins, a family of receptors that interacts to sialoglycans present in cell membranes, playing serving as immune checkpoints that regulate responses in cancer and other diseases **[31]**. In addition, *SIGLEC* genes are expressed in tumor cells and are associated with several parameters of aggressiveness [32]. It has been shown that SIGLECS plays important regulatory roles in the immune response by mediating cell-to-cell interactions, and they recognize the monosaccharide sialic acid on the surface of tumor cells. Cross-talk between distinct cell types and tumor environment seems to exist. In fact, the microenvironment promotes the abnormal secretion of sialic acid from tumor cells, which in turn stimulates the up-regulation of SIGLEC expression in infiltrating immune cells [33]. In hepatocellular carcinoma, *SIGLEC* family genes have potential prognostic value and may contribute to the regulation of cancer progression and immune cell infiltration [34]. Recently, Hou et al. (2022) found a high expression of *SIGLEC-15* in ATC and FTC. *SIGLEC-15* is indicated as a new immune checkpoint for thyroid cancer [35] since it acts as a putative oncogene, activating the STAT1/STAT3 signaling pathway to promote thyroid cancer cell growth, leading to an increase in immunosuppression [35].

The role of *SIGLEC10* and *SIGLEC11* genes in thyroid cancer is not yet investigated. In other cancer types, *SIGLEC10* is considered a pro-tumor marker and is associated with clinical aggressiveness [36,37]. Based on these observations altogether, a potential mechanism by which SIGLEC mRNA-loaded exosomes may increase thyroid cancer aggressiveness is depicted in Appendix A, and we are planning other studies to define the involvement of this gene on the molecular pathways of thyroid cancer. Regarding the *KRTAP* family, by using a genome-wide RNA interference screening to identify genes involved in cancer cell interaction with the endothelium, as well as vascular invasion and extravasation, *KRTAP5-5* emerged as a candidate gene. Its knockdown in breast cancer cells, causes vulnerability in invasive phenotypes [18]. 

The role of KRTAP proteins in thyroid cancer is not well characterized, for this reason, a possible future step of this study will be to investigate their role and function in the development and progression of thyroid cancer.

In conclusion, by combining transcriptomic and gene ontology analysis, a profile of de-regulated mRNAs present only in the exosomes released by PTC cells and mainly involved in epigenetic mechanism was identified. These preliminary data, could potentially help to identify new molecular targets for thyroid cancer. Further studies including functional assays and in vivo approaches will allow for the better characterization of the role of these de-regulated genes. Therefore, carrying on those studies, some exosomal mRNAs could be identified as important biomarkers for the diagnosis, monitoring and prognosis of PTCs with different aggressivity.

## 4. Materials and Methods

### 4.1. Cell Culture

In this study we used the human non-tumorigenic thyroid cell line Nthy-ori 3.1 (European Collection of Authenticated Cell Culture), which is widely adopted as a model of normal human thyroid cells [38], and a human PTC cell line, TPC-1, provided by Prof. A Fusco (University of Naples), which is characterized by the RET/PTC1 rearrangement [39]. Nthy-ori 3.1 and TPC-1 cells were cultured with RPMI 1640 (Thermo Fisher Scientific Inc., Waltham, MA, USA; code 61870) and DMEM (Thermo Fisher Scientific Inc.; code 31966) medium, respectively. The media were supplemented with 10% (*v/v*) heat-inactivated fetal bovine serum (Thermo Fisher Scientific Inc., Waltham, MA, USA; code 10500064), penicillin (100 IU/mL), streptomycin (0.1 mg/mL) (Thermo Fisher Scientific Inc., Waltham, MA, USA; code 15140-122) and amphotericin B (2.5 µg/mL) (Thermo Fisher Scientific Inc., Waltham, MA, USA; code 30-003). The cells were grown in a monolayer at 37 °C in 5% CO_2_ and humidified, as previously described [13]. To confirm the identity of the cell lines, short tandem repeat analysis was performed by using an AmpFLSTR NGM SElect PCR Amplification Kit (Thermo Fisher Scientific Inc., Waltham, MA, USA; code 4457889). 

### 4.2. Exosomes Isolation

The conditioned medium of TPC-1 and Nthy-ori 3.1 cells was collected, and exosomes were obtained using ExoQuick-TC (Systems Bioscience, Palo Alto, CA, USA; code EXOTC50A-1) according to manufacturer’s instructions. Briefly, TPC-1 (5 × 10^6^) and Nthy-ori 3.1 (4 × 10^6^) were seeded in 75 cm^2^ culture flasks. The next day, the growth medium was replaced with fresh medium supplemented with 10% exosome-depleted fetal bovine serum (Thermo Fisher Scientific Inc. Waltham, MA, USA; code A2720803) for 48 h [40]. After this time period, both cell lines reached 90% confluence. To remove cellular debris, the conditioned medium was collected, centrifuged at 4000 rpm for 30 min and filtered. Then, the exosomes were precipitated overnight at 4 °C using ExoQuick-TC (Systems Bioscience, Palo Alto, CA, USA). The next day, the mixture was centrifuged at 3000 rpm for 1 h and exosomes appeared as a beige pellet that it was resuspended in phosphate-buffered saline for DLS analysis, in lysis buffer for Western blotting analysis or stored at −80 °C until use for RNA extraction [13]. DLS and Western blotting analysis were performed to characterize the exosomes. In particular, the protein expression of CD63 and calregulin, classicals positive and negative hallmark of exosomes, respectively, were analyzed by Western blotting analysis [13].

### 4.3. RNA Extraction

To isolate total RNA from the thyroid cell culture samples and from the purified exosomes isolated from the conditioned media, TRIzol reagent (Thermo Fisher Scientific Inc., Waltham, MA, USA; code 15596026) and Total Exosome RNA (Thermo Fisher Scientific Inc., Waltham, MA, USA; code 4478545) were used [13]. RNA levels were quantified using a NanoDrop 2000 spectrophotometer (Thermo Fisher Scientific Inc., Waltham, MA, USA) and are as follows: 2.9 ng/µL and 5.8 ng/µL for TPC-1 and Nthy-ori 3.1, respectively.

### 4.4. Gene Expression Profiling 

Gene expression was evaluated using an AmpliSeq™ Transcriptome Human Gene Expression Kit (Thermo Fisher Scientific Inc.; code A26326), which is widely used in literature to assess gene expression variations [41,42,43]. Briefly, 5 ng of total RNA was reverse transcribed using a SuperScript™ IV VILO™ Master Mix (Thermo Fisher Scientific Inc., Waltham, MA, USA; code 11756050). The resulting cDNA was amplified using an Ion AmpliSeq™ Library Kit Plus and an Ion AmpliSeq Transcriptome Human Gene Expression core panel. Amplicons were digested, and barcoded adapters were ligated onto the target amplicons following manufacturer’s procedures. Libraries were equalized using an Ion Equalizer kit (Thermo Fisher Scientific Inc., Waltham, MA, USA; code 4482298) and eluted at 100 pM. Individual libraries were quantified using a Qubit 1× dsDNA HS kit (Thermo Fisher Scientific Inc., Waltham, MA, USA; code Q33230). Libraries were diluted to a 50 pM concentration and then combined in batches for further processing. Pooled libraries were templated on Ion Chef and sequenced using a 540 chip on the Ion GeneStudio S5 system. All samples were run in triplicate (technical replicate).

### 4.5. Data Analysis 

Ion Torrent Suite v5.12.3 software was used to map reads. Primary analysis for the AmpliSeq sequencing data of all samples was performed using the ampliSeqRNA plugin. This plugin uses the Torrent Mapping Alignment Program (TMAP) for aligning the raw sequencing reads against a custom reference sequence set (hg19_AmpliSeq_Transcriptome_21K_v1) containing all transcripts targeted by the kit. Differential gene expression analysis was performed using Transcriptome Analysis Console (TAC) Software version 4.0 (Appendix A). TAC uses the Limma Bioconductor package to analyze expression data. Student’s t test was used to compare the differential gene expression between cells and exosomes. Differentially expressed genes (DEGs) were selected based on an adjusted *p*-value (FDR *p*-value ≤ 0.05) and log2-fold change (log2(FC) ≤ −4 or ≥4). DEGs with RPM ≤ 10 in both comparison groups were omitted. Gene ontology analysis was performed using the Gene Ontology Enrichment Analysis and Visualization (GORILLA) tool.

## 5. Conclusions

Exosomes are actively released by cancer cells, and they favor tumor progression, mostly acting on the microenvironment. In carcinogenesis, exosomes have been proposed as efficient messengers to carry genetic and proteomic information between neighboring cells and distant organs. The main finding of the present study is the profile delineation of differentially present mRNAs among cell bodies and exosomes released by a tumorigenic and a non-tumorigenic cell line. In particular, in papillary thyroid cancer cell-derived exosomes, we found the up-regulation of SIGLEC and KRTAP mRNAs. These genes are involved in the malignant progression toward metastasis in other cancers. SIGLEC mRNAs present in exosomes could modulate functions of immune system cells. The molecular profiling of cancer exosomes would open new areas for monitoring events occurring in cancer microenvironments, and shed new light on mechanisms of thyroid tumorigenesis and help devise innovative therapeutic approaches. 

## Figures and Tables

**Figure 1 ijms-25-01407-f001:**
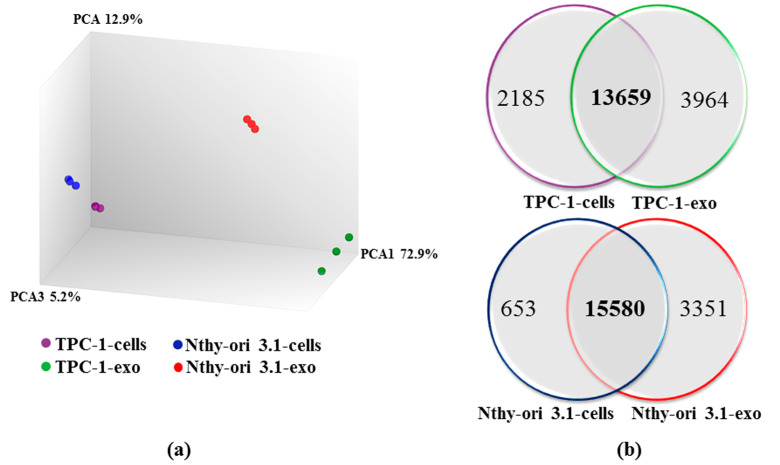
Analysis of the sequencing data. (**a**) Principal component analysis (PCA) results. Each point represent an RNA-seq sample. Samples with similar gene expression profiles are clustered together. (**b**) Venn diagrams represent the number of transcripts detected in TPC-1 cells (TPC-1-cells) and exosomes (TPC-1-exo) (upper panel), in Nthy-ori 3.1-cells and Nthy-ori 3.1-exo (lower panel), and that are common between the two samples of each cell line.

**Figure 2 ijms-25-01407-f002:**
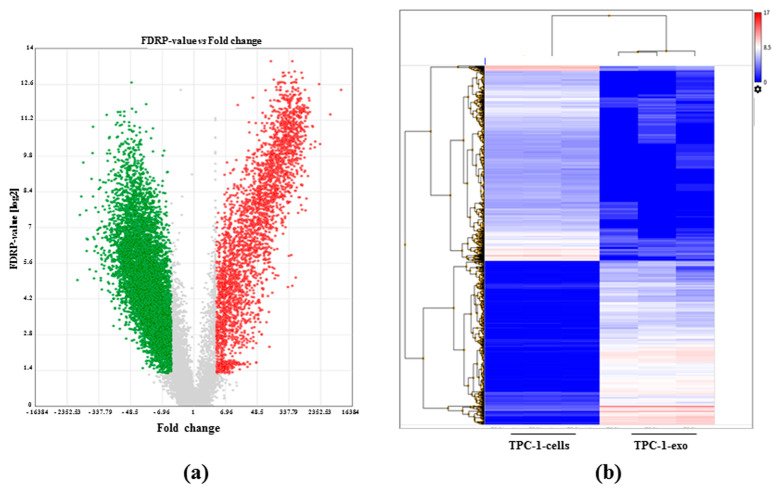
Differential gene expression between TPC-1 exosomes and cells. (**a**) Volcano plot summarizing significantly enriched transcripts. X-axis represents fold change (linear), and y-axis represent FDR *p*-value. Grey dots depict fold change- or *p*-value-based filtered transcripts. Green dots represent down-regulated transcripts between TPC-1-cells and TPC-1-exosomes (TPC-1-exo). Red dots represent up-regulated transcripts between TPC-1-cells and TPC-1-exo. (**b**) Heatmap representing the hierarchical clustering between gene expression between TPC-1-cells and TPC-1-exo obtained in triplicate analysis. Low-expressed transcripts are depicted in blue, while highly expressed ones are in red (data are expressed as log2 RPM).

**Figure 3 ijms-25-01407-f003:**
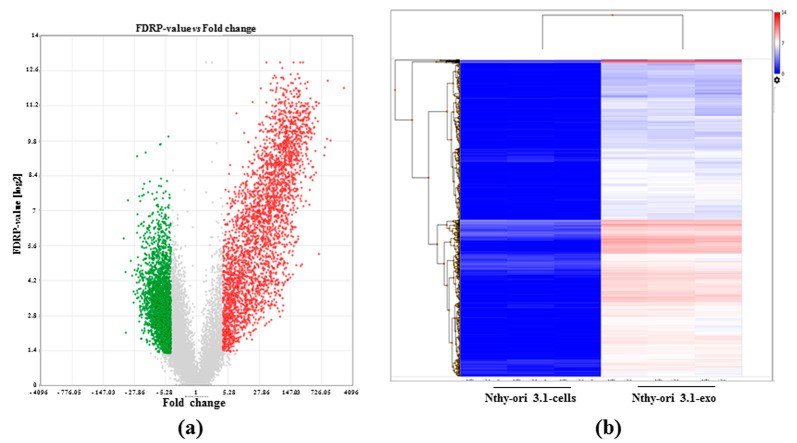
Differential gene expression between NThy-ori 3.1 exosomes and cells. (**a**) Volcano plot summarizing significantly enriched transcripts. X-axis represents fold change (linear), and y-axisrepresents FDR *p*-value. Grey dots depict fold change- or *p*-value-based filtered transcripts. Green dots represent down-regulated transcripts between NThy-ori 3.1-cells and NThy-ori 3.1 exosomes (NThy-ori 3.1-exo). Red dots represent up-regulated transcripts between NThy-ori 3.1-cells and NThy-ori 3.1-exo. (**b**) Heatmap representing the hierarchical clustering of gene expression between NThy-ori 3.1-cells and NThy-ori 3.1-exo obtained in triplicate analysis. Low-expressed transcripts are depicted in blue, while highly expressed ones are in red (data are expressed as log_2_ RPM).

**Figure 4 ijms-25-01407-f004:**
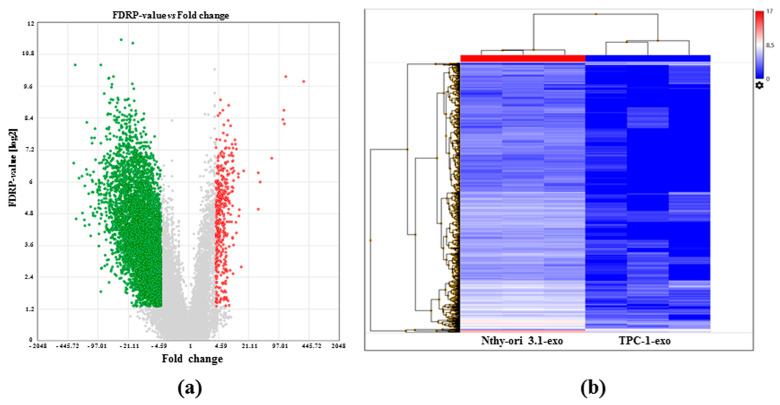
Differential gene expression between TPC1 and NThy-ori 3.1 exosomes. (**a**) Volcano plot summarizing significantly enriched transcripts. X-axis represents fold change (linear), and y-axis represent FDR *p*-value. Grey dots depict fold change- or *p*-value-based filtered transcripts. Green dots represent down-regulated transcripts between TPC-1 and NThy-ori 3.1 exosomes (TPC-1-exo, NThy-ori-exo). Red dots represent up-regulated transcripts between TPC-1 and NThy-ori 3.1. (**b**) Heatmap representing the hierarchical clustering between gene expression between TPC-1 and NThy-ori 3.1 exosomes. Low-expressed transcripts are depicted in blue, while highly expressed ones are in red (data are expressed as log2 RPM).

**Table 1 ijms-25-01407-t001:** Top 20 de-regulated transcripts in TPC1 exosomes compared to TPC1 cells.

UP-REGULATED				DOWN-REGULATED			
ID	Fold Change Log2 (FC)	*p*-Value	FDR *p*-value	ID	Fold Change Log2 (FC)	*p*-Value	FDR *p*-Value
**REXO1L2P**	13.00	6.74 × 10^−16^	4.01 × 10^−13^	**SPARC**	−10.29	3.07 × 10^−6^	1.15 × 10^−5^
**LOC440173**	12.08	3.11 × 10^−14^	3.74 × 10^−12^	**MMP2**	−10.07	3.21 × 10^−9^	3.23 × 10^−8^
**USP17L5**	11.17	1.34 × 10^−12^	5.26 × 10^−11^	**PSAP**	−9.94	4.39 × 10^−10^	6.01 × 10^−9^
**USP17L6P**	11.08	2.96 × 10^−16^	2.37 × 10^−13^	**ITM2B**	−9.76	1.13 × 10^−11^	2.85 × 10^−10^
**OR4M2**	10.71	1.03 × 10^−12^	4.27 × 10^−11^	**FN1**	−9.54	1.98 × 10^−9^	2.12 × 10^−8^
**PRM3**	10.70	1.97 × 10^−12^	7.15 × 10^−11^	**CLU**	−9.50	3.74 × 10^−8^	2.67 × 10^−7^
**ACTR3BP2**	10.51	1.63 × 10^−13^	1.11 × 10^−11^	**BSG**	−9.47	1.40 × 10^−7^	8.21 × 10^−7^
**SIGLEC10**	10.47	1.49 × 10^−13^	1.04 × 10^−11^	**LAMB3**	−9.38	4.16 × 10^−8^	2.92 × 10^−7^
**OR11H12**	10.36	1.41 × 10^−12^	5.45 × 10^−11^	**ATP1A1**	−9.21	9.68 × 10^−7^	4.28 × 10^−6^
**KRTAP19-2**	10.31	2.24 × 10^−12^	7.89 × 10^−11^	**HSP90B1**	−9.05	6.70 × 10^−11^	1.23 × 10^−9^
**SIGLEC11**	10.23	2.14 × 10^−14^	2.96 × 10^−12^	**VIM**	−8.97	3.75 × 10^−12^	1.19 × 10^−10^
**KRTAP5-3**	10.22	4.03 × 10^−15^	1.04 × 10^−12^	**SPP1**	−8.91	4.42 × 10^−10^	6.04 × 10^−9^
**OR6X1**	10.18	2.64 × 10^−15^	8.41 × 10^−13^	**APP**	−8.90	1.65 × 10^−13^	1.12 × 10^−11^
**HSPA7**	10.10	4.87 × 10^−11^	9.41 × 10^−10^	**FKBP10**	−8.88	3.23 × 10^−8^	2.36 × 10^−7^
**MRGPRD**	10.09	2.87 × 10^−14^	3.52 × 10^−12^	**ITGB3**	−8.82	2.52 × 10^−6^	9.68 × 10^−6^
**KRTAP10-11**	10.09	1.09 × 10^−14^	2.02 × 10^−12^	**CTSD**	−8.80	3.90 × 10^−8^	2.77 × 10^−7^
**KRBOX1**	10.06	1.69 × 10^−13^	1.13 × 10^−11^	**HLA-E**	−8.79	1.40 × 10^−7^	8.20 × 10^−7^
**OR2T2**	10.03	2.34 × 10^−14^	3.06 × 10^−12^	**ARF5**	−8.71	1.45 × 10^−9^	1.64 × 10^−8^
**MIR548I1**	9.99	1.02 × 10^−14^	1.94 × 10^−12^	**HNRPDL**	−8.68	1.23 × 10^−6^	5.26 × 10^−6^
**KRTAP5-8**	9.94	5.58 × 10^−10^	7.37 × 10^−9^	**PDIA6**	−8.66	9.04 × 10^−8^	5.64 × 10^−7^

**Table 2 ijms-25-01407-t002:** Top 20 de-regulated transcripts in Nthy-ori 3.1 exosomes compared to Nthy-ori 3.1 cells.

UP-REGULATED				DOWN-REGULATED			
ID	Fold Change Log2 (FC)	*p*-Value	FDR *p*-Value	ID	Fold Change Log2 (FC)	*p*-Value	FDR *p*-Value
**REXO1L2P**	11.32	2.44 × 10^−15^	1.24 × 10^−12^	**COL18A1**	−5.62	1.17 × 10^−7^	1.30 × 10^−6^
**USP17L5**	10.30	2.98 × 10^−12^	1.54 × 10^−10^	**LAMA5**	−5.49	3.85 × 10^−6^	2.88 × 10^−5^
**USP17L6P**	10.07	9.08 × 10^−16^	6.30 × 10^−13^	**RTTN**	−5.45	2.8 × 10^−3^	7.6 × 10^−3^
**LOC349196**	10.03	2.48 × 10^−12^	1.33 × 10^−10^	**CTGF**	−529	2.22 × 10^−9^	3.83 × 10^−8^
**PRM3**	9.87	6.72 × 10^−12^	2.87 × 10^−10^	**PHTF2**	−5.08	1.20 × 10^−6^	1.03 × 10^−5^
**DUX4L4**	9.81	1.73 × 10^−11^	6.16 × 10^−10^	**B4GALNT4**	−4.98	5.61 × 10^−6^	3.98 × 10^−5^
**KRTAP10-8**	9.40	2.07 × 10^−14^	4.90 × 10^−12^	**ANKRD17**	−4.87	2.99 × 10^−5^	2 × 10^−4^
**KRT16P3**	9.39	5.82 × 10^−07^	5.44 × 10^−6^	**NRP1**	−4.86	1 × 10^−4^	5 × 10^−4^
**REXO1L1**	9.32	2.85 × 10^−13^	2.62 × 10^−11^	**SPPL2B**	−4.77	3.03 × 10^−6^	2.34 × 10^−5^
**LOC339240**	9.21	1.61 × 10^−14^	4.41 × 10^−12^	**RAI1**	−4.72	4.10 × 10^−5^	2 × 10^−4^
**KRTAP10-11**	9.20	2.93 × 10^−14^	6.16 × 10^12^	**BIRC6**	−4.67	5.02 × 10^−6^	3.62 × 10^−5^
**GOLGA6L1**	9.17	1.13 × 10^−10^	2.95 × 10^−9^	**ANKRD36**	−4.61	1.14 × 10^−5^	7.38 × 10^−5^
**RXFP3**	9.15	1.72 × 10^−14^	4.47 × 10^−12^	**ZCCHC11**	−4.61	1.83 × 10^−11^	6.44 × 10^−10^
**KRTAP19-2**	9.13	1.02 × 10^−11^	4.03 × 10^−10^	**NID2**	−4.60	3.30 × 10^6^	2.52 × 10^−5^
**OR2B11**	9.12	1.80 × 10^−13^	1.98 × 10^−11^	**ARGLU1**	−4.60	6.52 × 10^−8^	7.64 × 10^−7^
**LOC100240734**	9.05	1.39 × 10^−12^	8.26 × 10^−11^	**PHF2**	−4.59	6.26 × 10^−5^	3 × 10^−4^
**LOC440173**	9.04	5.76 × 10^−13^	4.45 × 10^−11^	**PCNXL3**	−4.58	2.77 × 10^−5^	2 × 10^−4^
**HTR1A**	9.02	4.78 × 10^−14^	8.37 × 10^−12^	**NUP210**	−4.58	3.17 × 10^−6^	2.44 × 10^−5^
**LOC285548**	9.02	6.24 × 10^−13^	4.77 × 10^−11^	**MKL2**	−4.57	9.51 × 10^−9^	1.39 × 10^−7^
**HSPA7**	9.01	1.50 × 10^−10^	3.72 × 10^−9^	**SREBF1**	−4.56	3.51 × 10^−5^	2 × 10^−4^

## Data Availability

Data is contained within the article and Appendix A.

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
