# Peer review of "RNA Profile of Cell Bodies and Exosomes Released by Tumorigenic and Non-Tumorigenic Thyroid Cells"

_ijms, 2024, doi:10.3390/ijms25031407_

Round 1
Reviewer 1 Report
Comments and Suggestions for Authors
This study compared the gene expression of exosomes and the corresponding cells in non-tumorigenic thyroid cell line, and in papillary thyroid cancer. The text is well written, and it is easy to read and understand. I am not sure, but it seems the comparison of exosomes between non-tumor and cancer is not shown. The comparison of cell bodies between the non-tumor and cancer is not shown as well. If not done, the manuscript may benefit from data reanalysis.
Additional comments:
(1) Line 311, regarding "Cells were cultured in RPMI or DMEM (Thermo Fisher Scientific Inc., Waltham, MA, USA) media at 37° C in a humidified 5% CO2 atmosphere, as previously described [13]."
Could you please provide the details?
(2) Please add the catalog numbers of the reagents that were used and are detailed in the materials and methods.
(3) Regarding section 4.3. As I understand, only the RNA present inside the exosomes was analyzed?
(4) Regarding the Ion AmpliSeq™ Transcriptome Human Gene Expression Kit. Could you please specify that it is sequencing analysis? Apart from the gene expression data, was RNA sequencing data available as well (so you have information of mutational, copy-number, etc. )?
(5) Line 348. Regarding "All samples were run in triplicate (technical replicate).". As I understand, the reading of the gene expression profiling was done 3 times. Was the signal for each gene-probe averaged?
(6) Since it is run in triplicate. Was the cell culture, exosome isolation and RNA extraction also in triplicate? Or just the trasncriptome analysis?
(7) Line 358, regarding "One way ANOVA with post-hoc Tukey was used for RNA sequencing analysis". Since you are comparing 2 groups, why do you need post-hoc analysis? Or in the analysis the 4 groups are compared?
(8) Could you please add a figure with the methodology depicted?
(9) In the methodology, first the exosomes are isolated, and later the remaining cells were also extracted/isolated? So it is a 2x2 table?
(10) Why there is no figure of the comparison between TPC-1 and Nthy-ori 3.1 in terms of cells or exosomes?
(11) Do you have an image of the western blot of CD63 and caregulin to be included in the manuscript?
(12) Could you please upload the Excel or text file with the gene expression values that you can obtain from the TAC software? Please add this data in the supplementary data.
(13) In the output shown in Table 1 and 2, is there any micro RNA present?
(14) Were SIGLEC genes upregulated in both neoplastic and non-neoplastic cells?
(15) Can you show the comparison of exosomes between normal and neoplastic cells?
(16) You may compare you data with publicly available gene expression data of thyroid cancer.
Author Response
Reviewer 1 (Comments/criticisms of the Reviewers are reported in Italics)
English language fine. No issues detected
This study compared the gene expression of exosomes and the corresponding cells in non-tumorigenic thyroid cell line, and in papillary thyroid cancer. The text is well written, and it is easy to read and understand. I am not sure, but it seems the comparison of exosomes between non-tumor and cancer is not shown. The comparison of cell bodies between the non-tumor and cancer is not shown as well. If not done, the manuscript may benefit from data reanalysis.
Additional comments:
(1) Line 311, regarding "Cells were cultured in RPMI or DMEM (Thermo Fisher Scientific Inc., Waltham, MA, USA) media at 37° C in a humidified 5% CO2 atmosphere, as previously described [13]."
Could you please provide the details?
Answer: Nthy-ori 3.1 and TPC-1 cells were cultured with RPMI 1640 and DMEM medium, respectively. The media were supplemented with 10% (v/v) heat-inactivated fetal bovine serum, penicillin (100 IU/ml), streptomycin (0.1mg/ml) and amphotericin B (2.5 µg/ml). The cells were grown in monolayer at 37°C in 5% CO2 humidified atmosphere. Now we added in the section Materials and Methods this information (page 8, lines 313-318)
(2) Please add the catalog numbers of the reagents that were used and are detailed in the materials and methods.
Answer: Done. We added the code in the section Materials and Methods.
(3) Regarding section 4.3. As I understand, only the RNA present inside the exosomes was analyzed?
Answer: In the study we analyzed both the total RNA isolated from thyroid cells and purified exosomes (page 9, line 342)
(4) Regarding the Ion AmpliSeq™ Transcriptome Human Gene Expression Kit. Could you please specify that it is sequencing analysis? Apart from the gene expression data, was RNA sequencing data available as well (so you have information of mutational, copy-number, etc. )?
Answer: The Ion AmpliSeq™ Transcriptome Human Gene Expression Kit is a precast sequencing kit that enables the detection of the expression levels of > 95% human RefSeq genes. Amplicon-based sequencing is performed using primers targeting 18,574 mRNAs and 2,228 non-coding RNAs. This kit allows only gene expression analysis and no SNV detection or alternative splice site variants. This kit is widely used in literature to assess gene expression variations as now indicated in the section Materials and Methods (page 9,lines 351-352) (new references 41-43).
(5) Line 348. Regarding "All samples were run in triplicate (technical replicate).". As I understand, the reading of the gene expression profiling was done 3 times. Was the signal for each gene-probe averaged?
Answer: The Transcriptome Analysis Console (TAC) Software was used to assess gene expression data. Signals form the three technical replicates were averaged and standard deviations were calculated. After comparison, signals with an adjusted p value < 0.05 was considered. It should be considered that the sequencing kit allows the amplification of a single probe for each transcript analyzed.
(6) Since it is run in triplicate. Was the cell culture, exosome isolation and RNA extraction also in triplicate? Or just the transcriptome analysis?
Answer: We don’t performed exosome isolation and RNA extraction in triplicate but the exctracted the samples were run in triplicate in transciptome analysis.
(7) Line 358, regarding "One way ANOVA with post-hoc Tukey was used for RNA sequencing analysis". Since you are comparing 2 groups, why do you need post-hoc analysis? Or in the analysis the 4 groups are compared?
Answer: Thanks for your comment. Following this we noted the presence of our mistake in this part of method. Indeed Student’s t test was used to compare differential gene expression between cells and exosomes. The manuscript has been updated accordingly (page 9, lines 373-374)
(8) Could you please add a figure with the methodology depicted?
Answer: Done. We added in the manuscript a new supplementary Figure 1 (page 7 line 243-244)
(9) In the methodology, first the exosomes are isolated, and later the remaining cells were also extracted/isolated? So it is a 2x2 table?
Answer: We isolated the exosomes by culture medium and meanwhile isolated total RNA from cultured cells, so in the study there are two conditions for each cell lines (exosomal RNA and cell bodies RNA)
(10) Why there is no figure of the comparison between TPC-1 and Nthy-ori 3.1 in terms of cells or exosomes?
Answer: “In panel A of figure 1, the PCA shows the difference between the conditions investigated and clearly indicates that differences between the two cell bodies are lower then differences between cell bodies and exosomes. The focus of our research is to detect differences between cell bodies and exosomes; therefore we believe that panel A of Figure 1 is sufficient for the reader”. The comparison between TPC-1 and Nthy-ori 3.1 exosomes is shown in figure 4.
(11) Do you have an image of the western blot of CD63 and caregulin to be included in the manuscript?
Answer: The western blot image of CD63 and calregulin, related to the isolated exosomes used for the transciptome analysis in the current work, it has already been published in a previous article (reference n.13) as reported in the results section (page 2, lines 74-75).
(12) Could you please upload the Excel or text file with the gene expression values that you can obtain from the TAC software? Please add this data in the supplementary data.
Answer: The complete list of gene expression data are now available as supplementary table 3
(13) In the output shown in Table 1 and 2, is there any micro RNA present?
Answer: In table 1 there is the miR-548I1. In the Human microRNA Disease Database this miR is associated with colorectal, non small cell lung and hepatocellular carcinomas. We decided do not consider the miRNA because we focused our attention only on long noncoding and coding RNA genes.
(14) Were SIGLEC genes upregulated in both neoplastic and non-neoplastic cells?
Answer: Siglec genes are de-regulated in tumorigenic and non-tumorigenic exosomes and as shown in table 1 their expression is higher in tumoral exosomes.
(15) Can you show the comparison of exosomes between normal and neoplastic cells?
Answer: In the manuscript, the comparison between non-tumorigenic and tumorigenic exosomes is shown in figure 4.
(16) You may compare you data with publicly available gene expression data of thyroid cancer.
Answer: Several studies delineated gene expression profile of thyroid cancer cells (i.e. TGCA studies), while in this work for the first time, we investigated and compared the RNA profile of cell bodies and exosomes released by tumorigenic and non-tumorigenic thyroid cells.
Reviewer 2 Report
Comments and Suggestions for Authors
The manuscript represents a genetic analysis of exosomes from thyroid tumors. In principle, the results are interesting although expected. There are several points which were not clearly stated in the text and should be pointed out. The positive identification of exosomes with specific markers was not performed, the protocol does not provide a sufficient detail required. There are several pathways which seem contradictory for example HLA-E, ITM2B found in the exosomes. The difference is part of the exosome data may be the culture conditions. The authors do not seem to address the issue of culture manipulation since there is a huge difference by using DMEM as compare to RPMI based on the glucose content. It would be interesting to see the pattern in serum free media and glucose free media. Finally, there should be a clear difference in growth rate between the cell lines which may influence the content in the exosomes.
Comments on the Quality of English LanguageThere are minor gramatical mistakes in the text.
Author Response
Reviewer 2 (Comments/criticisms of the Reviewers are reported in Italics)
The manuscript represents a genetic analysis of exosomes from thyroid tumors. In principle, the results are interesting although expected. There are several points which were not clearly stated in the text and should be pointed out.
- The positive identification of exosomes with specific markers was not performed, the protocol does not provide a sufficient detail required.
Answer: The western blot image of CD63 and calregulin, related to the isolated exosomes used for the transciptome analysis in the current work, it has already been published in a previous article (reference n.13) as reported in the results section (page 2, lines 74-75). Now, as suggested, we better specify in the Materials and Methods, the details on the characterization of exosomes (page 9, lines 336-339)
- There are several pathways which seem contradictory for example HLA-E, ITM2B found in the exosomes. The difference is part of the exosome data may be the culture conditions. The authors do not seem to address the issue of culture manipulation since there is a huge difference by using DMEM as compare to RPMI based on the glucose content. It would be interesting to see the pattern in serum free media and glucose free media. Finally, there should be a clear difference in growth rate between the cell lines which may influence the content in the exosomes.
Answer: The culture medium composition and preparation are important factors when in vitro study with cell culture studies are performed. For this reason, in our work, as also reported in several published works and also suggested in MISEV 2018 (new reference 40) we maintained the cells with medium supplemented with 10% exosome-depleted fetal bovine serum (as indicated in the section materials and methods page 8 line 328-329) before the exosomes extraction. Moreover, we want clarify that, in our study the content in the exosomes cannot be influenced by difference in growth rate between the cell lines because the cells were seeded in 75 cm2 culture flasks, so both cells had reached 90% confluence when the conditioned medium was taken to extract the exosomes. We specify this detail in the section Material and Methods (page 9, lines 329-330) and we added in the manuscript a schematic protocol of the samples processing of our study (new supplementary Figure 1 ; page 7 lines 243-244)
- Minor editing of English language required
Answer: Editing of English language done.
Round 2
Reviewer 2 Report
Comments and Suggestions for Authors
The manuscript was partially improved and may be considered for publication
Comments on the Quality of English LanguageMinor grammatical mistakes were encountered